# Modular Flows: Differential Molecular Generation

**Yogesh Verma, Samuel Kaski, Markus Heinonen**
Aalto University
{yogesh.verma, samuel.kaski, markus.heinonen}@aalto.fi

**Vikas Garg**
YaiYai Ltd and Aalto University
vgarg@csail.mit.edu; vikas@yaiyai.fi

## Abstract

Generating new molecules is fundamental to advancing critical applications such as drug discovery and material synthesis. Flows can generate molecules effectively by inverting the encoding process, however, existing flow models either require artifactual dequantization or specific node/edge orderings, lack desiderata such as permutation invariance, or induce discrepancy between the encoding and the decoding steps that necessitates *post hoc* validity correction. We circumvent these issues with novel continuous normalizing E(3)-equivariant flows, based on a system of node ODEs coupled as a graph PDE, that repeatedly reconcile locally toward globally aligned densities. Our models can be cast as message passing temporal networks, and result in superlative performance on the tasks of density estimation and molecular generation. In particular, our generated samples achieve state of the art on both the standard QM9 and ZINC250K benchmarks.

## 1 Introduction

Generative models have rapidly become ubiquitous in machine learning with advances from image synthesis (Ramesh et al., 2022) to protein design (Ingraham et al., 2019). Molecular generation (Stokes et al., 2020) has also received significant attention owing to its promise for discovering new drugs and materials. Searching for valid molecules in prohibitively large discrete spaces is, however, challenging: estimates for drug-like structures range between $10^{23}$ and $10^{60}$ but only a tiny fraction - on the order of $10^8$ - has been synthesized (Polishchuk et al., 2013; Merz et al., 2020). Thus, learning representations that exploit appropriate molecular inductive biases (e.g., spatial correlations) becomes crucial.

Earlier models focused on generating sequences based on the SMILES notation (Weininger, 1988) used in Chemistry to describe the molecular structures as strings. However, they were supplanted by genera-tive models that capture valuable spatial information such as bond strengths and dihedral angles, e.g., by embedding molecular graphs via some graph neural network (GNNs) (Scarselli et al., 2009; Garg et al., 2020). Such models primarily include variants of Generative Adversarial Networks (GANs), Variational Autoencoders (VAEs), and Normalizing

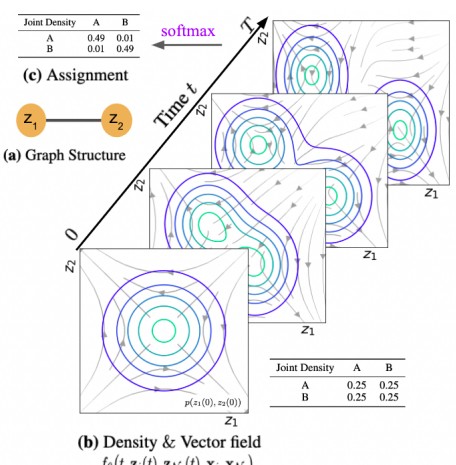

Figure 1: A toy illustration of `ModFlow` in action with a two-node graph. The two local flows - $\mathbf{z}_1$ and $\mathbf{z}_2$ - co-evolve toward a more complex joint density, both driven by the same differential $f$.

36th Conference on Neural Information Processing Systems (NeurIPS 2022).

Flows (Dinh et al., 2014, 2016). Besides known issues with their training, GANs (Goodfellow et al., 2014; Maziarka et al., 2020) suffer from the well-documented problem of mode collapse, thereby generating molecules that lack diversity. VAEs (Kingma and Welling, 2013; Lim et al., 2018; Jin et al., 2018), on the other hand, are susceptible to a distributional shift between the training data and the generated samples. Moreover, optimizing for likelihood via a surrogate lower bound is likely insufficient to capture the complex dependencies inherent in the molecules.

Flows are especially appealing since, in principle, they enable estimating (and sampling from) complex data distributions using a sequence of invertible transformations on samples from a more tractable continuous distribution. Molecules are discrete, so many flow models (Madhawa et al., 2019; Honda et al., 2019; Shi et al., 2020) add noise during encoding and later apply a *dequantization* procedure. However, dequantization begets distortion and issues related to convergence (Luo et al., 2021). Moroever, many methods segregate the generation of atoms from bonds, so the decoded structure is often not a valid molecule and requires *post hoc* correction to ensure validity (Zang and Wang, 2020), effecting a discrepancy between the encoding and the decoded distributions. Permutation dependence is another undesirable artifact of these methods. Some alternatives have been explored to avoid dequantization, e.g., (Lippe and Gavves, 2021) encodes molecules in a continuous latent space via variational inference and jointly optimizes a flow model for generation. Discrete graph flows (Luo et al., 2021) also circumvent the many pitfalls of dequantization by resorting to discrete latent variables, and performing validity checks during the generative process. However, discrete flows follow an autoregressive procedure that requires a specific ordering of nodes and edges during training. In general, one shot methods can generate much faster than discrete flows.

We offer a different flow-based perspective tailored to molecules. Specifically, we suggest coupled continuous normalizing E(3)-equivariant flows that bestow generative capabilities from neural partial differential equation (PDE) models on graphs. Graph PDEs have been known to enable designing new embedding methods such as variants of GNNs (Chamberlain et al., 2021), extending GNNs to continuous layers as Neural ODEs (Poli et al., 2019), and accommodating spatial information (Iakovlev et al., 2020). We instead seek to bring to the fore their efficacy and elegance as tools to help generate complex objects, such as molecules, viewed as outcomes resulting from an interplay of co-adapting latent trajectories (i.e., underlying dynamics). Concretely, a flow is associated with each node of the graph, and these flows are conjoined as a joint ODE system conditioned on neighboring nodes. While these flows originate independently as samples from simple distributions, they adjust progressively toward more complex joint distributions as they repeatedly interact with the neighboring flows. We view molecules as samples generated from the globally aligned distributions obtained after many such local feedback iterations. We call the proposed method Modular Flows (`ModFlows`) to underscore that each node can be regarded as a module that coordinates with other modules. Table 1 summarizes the capabilities of `ModFlow` compared to some previous generative works.

**Contributions.** We propose to learn continuous-time, flow based generative models, grounded on graph PDEs, for generating molecules without resorting to any validity correction. In particular,

- we propose `ModFlow`, a novel generative model based on coupled continuous normalizing E(3)-equivariant flows. `ModFlow` encapsulates essential inductive bias using PDEs, and defines multiple flows that interact locally toward a globally consistent joint density;

Table 1: A comparison of generative modeling approaches for molecules.

| Method | One-shot | Modular | Invertible | Continuous-time | |
|---|---|---|---|---|---|
| JT-VAE | ✓ | ✓ | ✗ | ✗ | Jin et al. (2018) |
| MRNN | ✗ | ✗ | ✗ | ✗ | Popova et al. (2019) |
| GraphAF | ✗ | ✗ | ✓ | ✗ | Shi et al. (2020) |
| GraphDF | ✗ | ✗ | ✓ | ✗ | Luo et al. (2021) |
| MoFlow | ✓ | ✗ | ✓ | ✗ | Zang and Wang (2020) |
| GraphNVP | ✓ | ✗ | ✓ | ✗ | Madhawa et al. (2019) |
| `ModFlow` | ✓ | ✓ | ✓ | ✓ | this work |

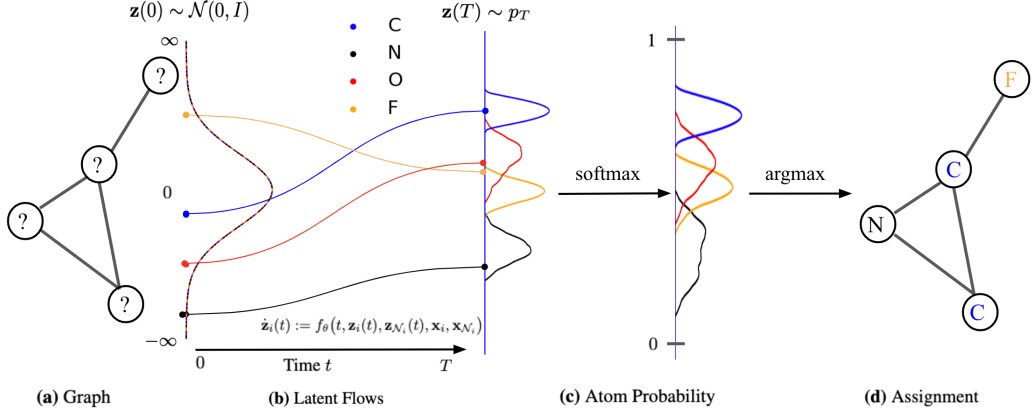

$\mathbf{z}(0) \sim \mathcal{N}(0, I)$      $\mathbf{z}(T) \sim p_T$

C
N
O
F

softmax      argmax

$\dot{\mathbf{z}}_i(t) := f_\theta\big(t, \mathbf{z}_i(t), \mathbf{z}_{\mathcal{N}_i}(t), \mathbf{x}_i, \mathbf{x}_{\mathcal{N}_i}\big)$

Time $t$

**(a)** Graph      **(b)** Latent Flows      **(c)** Atom Probability      **(d)** Assignment

Figure 2: A demonstration of the modular flow generation. The initial Gaussian distributions $\mathcal{N}(0, I)$ evolve into complex densities $\mathbf{z}(T)$ under $f$ and are subsequently translated into probabilities and labels.

- we encode permutation, translation, rotation, and reflection equivariance with E(3) equivariant GNNs adapted to molecular generation, and can leverage 3D geometric information;

- ModFlow is end-to-end trainable, non-autoregressive, and obviates the need for any external validity checks or correction;

- empirically, ModFlow achieves state-of-the-art performance on both the standard QM9 (Ramakrishnan et al., 2014) and ZINC250K (Irwin et al., 2012) benchmarks.

## 2 Related works

**Generative models.** Earlier attempts for molecule generation (Kusner et al., 2017; Dai et al., 2018) aimed at representing molecules as SMILES strings (Weininger, 1988) and developed sequence generation models. A challenge for these approaches is to learn complicated grammar rules that can generate syntactically valid sequences of molecules. Recently, representing molecules as graphs has inspired new deep generative models for molecular generation (Segler et al., 2018; Samanta et al., 2018; Neil et al., 2018), ranging from VAEs (Jin et al., 2018; Kajino, 2019) to flows (Madhawa et al., 2019; Luo et al., 2021; Shi et al., 2020). The core idea is to learn to encode molecular graphs into a latent space, and subsequently decode samples from this latent space to generate new molecules (Atwood and Towsley, 2016; Xhonneux et al., 2020; You et al., 2018).

**Graph partial differential equations.** Graph PDEs is an emerging area that studies PDEs on structured data encoded as graphs. For instance, one can define a PDE on graphs to track the evolution of signals defined over the graph nodes under some dynamics. Graph PDEs have enabled, among others, design of new graph neural networks; see, e.g., works such as GNODE (Poli et al., 2019), NeuralPDE (Iakovlev et al., 2020), Neural operator (Li et al., 2020), GRAND (Chamberlain et al., 2021), and PDE-GCN (Eliasof et al., 2021). Different from all these works, we focus on using PDEs for generative modeling of molecules (as graph-structured objects). Interestingly, ModFlow proposed in this work may be viewed as a new equivariant temporal graph network (Rossi et al., 2020; Souza et al., 2022).

**Validity oracles.** A key challenge of molecular generative models is to be able to generate valid molecules, according to various criteria for molecular validity or feasibility. It is a common practice to call on external chemical software as rejection oracles to reduce or exclude invalid molecules, or do validity checks as part of autoregressive generation (Luo et al., 2021; Shi et al., 2020; Popova et al., 2019). An important open question has been whether generative models can learn to achieve high generative validity *intrinsically*, i.e., without being aided by oracles or resorting to additional checks. ModFlow takes a major step forward toward that goal.

# 3 Modular Flows

We focus on unsupervised learning of an underlying graph density $p(G)$ using a dataset of observed molecular graphs $\mathcal{D} = \{G_n\}_{n=1}^N$. We learn a generative flow model $p_\theta(G)$ specified by flow parameters $\theta$, and use it to sample novel high-probability molecules.

## 3.1 Molecular Representation

**Graph representation.** We represent each molecular graph $G = (V, E)$ of size $M$ as a tuple of vertices $V = (v_1, \ldots, v_M)$ and edges $E \subset V \times V$. Each vertex takes a value from an alphabet on atoms: $v \in \mathcal{A} = \{\texttt{C}, \texttt{H}, \texttt{N}, \texttt{O}, \texttt{P}, \texttt{S}, \ldots\}$; while each edge $e \in \mathcal{B} = \{1, 2, 3\}$ abstracts some bond type (i.e., single, double, or triple). We assume that, conditioned on the edges, the graph likelihood factorizes as a product of categorical distributions over vertices given their latent representations:

$$p(G) := p(V|E, \{z\}) = \prod_{i=1}^M \text{Cat}(v_i|\sigma(\mathbf{z}_i)) \,, \tag{1}$$

where $\mathbf{z}_i = (z_{i\texttt{C}}, z_{i\texttt{H}}, \ldots) \in \mathbb{R}^{|\mathcal{A}|}$ is a set of atom scores for node $i$ such that $z_{ik} \in \mathbb{R}$ pertains to type $k \in \mathcal{A}$, and $\sigma$ is the softmax function

$$\sigma(\mathbf{z}_i)_k = \frac{\exp(\mathbf{z}_{ik})}{\sum_{k'} \exp(\mathbf{z}_{ik'})} \,, \tag{2}$$

which turns the real-valued scores $\mathbf{z}_i$ into normalized probabilities. `ModFlow` also supports 3D molecular graphs that contain atomic coordinates and angles as additional information.

**Tree representations.** We can obtain an alternative representation for molecules: we can decompose each molecule into a tree-like structure, by contracting certain vertices into a single node (denoted as a cluster) such that the molecular graph becomes acyclic. Following Jin et al. (2018), we restrict these clusters to ring substructures present in the molecular data, in addition to the atom alphabet. Thus, we obtain an extended alphabet $\mathcal{A}_{\text{tree}} = \mathcal{A} \cup \{\texttt{C}_1, \texttt{C}_2, \ldots\}$, where each cluster label $\texttt{C}_r$ corresponds to some ring substructure in the label vocabulary $\mathcal{X}$. We then reduce the vocabulary to the 30 most commonly occurring substructures of $\mathcal{A}_{\text{tree}}$. For further details, see Appendix A.2.

## 3.2 Differential modular flows

Normalizing flows (Kobyzev et al., 2021) provide a general recipe for constructing flexible probability distributions, used in density estimation (Cramer et al., 2021; Huang et al., 2018) and generative modeling (Zhen et al., 2020; Zang and Wang, 2020). We propose to model the atom scores $\mathbf{z}_i(t)$ as a Continuous-time Normalizing Flow (CNF) (Grathwohl et al., 2018) over time $t \in \mathbb{R}_+$. We assume the initial scores at time $t = 0$ follow an uninformative Gaussian base distribution $\mathbf{z}_i(0) \sim \mathcal{N}(0, I)$ for each node $i$. Node scores evolve in parallel over time according to the differential equation

$$\dot{\mathbf{z}}_i(t) := \frac{\partial \mathbf{z}_i(t)}{\partial t} = f_\theta\big(t, \mathbf{z}_i(t), \mathbf{z}_{\mathcal{N}_i}(t), \mathbf{x}_i, \mathbf{x}_{\mathcal{N}_i}\big), \qquad i \in \{1, \ldots, M\} \,, \tag{3}$$

where $\mathcal{N}_i = \{j : (i, j) \in E\}$ is the set of neighbors of node $i$ and $\mathbf{z}_{\mathcal{N}_i}(t) = \{\mathbf{z}_j(t) : j \in \mathcal{N}_i\}$ the scores of the neighbors at time $t$; $\mathbf{x}_i$ and $\mathbf{x}_{\mathcal{N}_i}$ denote, respectively, the positional (2D/3D) information of $i$ and its neighbours; and $\theta$ denotes the parameters of the flow function $f$ to be learned. Stacking together all node differentials, we obtain a *modular* system of coupled ODEs:

$$\dot{\mathbf{z}}(t) = \begin{pmatrix} \dot{\mathbf{z}}_1(t) \\ \vdots \\ \dot{\mathbf{z}}_M(t) \end{pmatrix} = \begin{pmatrix} f_\theta\big(t, \mathbf{z}_1(t), \mathbf{z}_{\mathcal{N}_1}(t), \mathbf{x}_1, \mathbf{x}_{\mathcal{N}_1}\big) \\ \vdots \\ f_\theta\big(t, \mathbf{z}_M(t), \mathbf{z}_{\mathcal{N}_M}(t), \mathbf{x}_M, \mathbf{x}_{\mathcal{N}_M}\big) \end{pmatrix} \tag{4}$$

$$\mathbf{z}(T) = \mathbf{z}(0) + \int_0^T \dot{\mathbf{z}}(t)dt \,. \tag{5}$$

This coupled system of ODEs may be viewed as a graph PDE (Iakovlev et al., 2020; Chamberlain et al., 2021), where the evolution of each node depends only on its neighbors.

The joint flow induces a corresponding change in the individual densities in terms of divergence of $f$ (Chen et al., 2018),

$$\frac{d \log p_t(\mathbf{z}_i(t))}{dt} = -\operatorname{tr}\left(\frac{\partial f_\theta\big(t, \mathbf{z}_i(t), \mathbf{z}_{\mathcal{N}_i}(t), \mathbf{x}_i, \mathbf{x}_{\mathcal{N}_i}\big)}{\partial \mathbf{z}_i}\right), \tag{6}$$

starting from the base distribution $p_0(\mathbf{z}_i(0)) = \mathcal{N}(\mathbf{z}_i(0)|0, I)$. The trace picks only the diagonal elements of the Jacobian $\frac{\partial f}{\partial \mathbf{z}}$, which interprets the input from neighbors, $\mathbf{z}_{\mathcal{N}_i}$, as a 'control' for each node $\mathbf{z}_i$ at each instant $t$. An ODE solver is used for such systems, and the gradients are computed via the adjoint sensitivity method (Kolmogorov et al., 1962). This approach incurs a low memory cost, and explicitly controls the numerical error. Notably, moving towards modular flows translates sparsity also to the adjoints.

**Proposition 1:** *Modular adjoints are sparser than regular adjoints. They can be computed as*

$$\frac{d\boldsymbol{\lambda}_i}{dt} = -\boldsymbol{\lambda}(t)^\top \frac{\partial f\big(t, \mathbf{z}_i(t), \mathbf{z}_{\mathcal{N}_i}(t), \mathbf{x}_i, \mathbf{x}_{\mathcal{N}_i}\big)}{\partial \mathbf{z}} = -\sum_{j \in \mathcal{N}_i \cup \{i\}} \boldsymbol{\lambda}_j(t)^\top \frac{\partial f\big(t, \mathbf{z}_i(t), \mathbf{z}_{\mathcal{N}_i}(t), \mathbf{x}_i, \mathbf{x}_{\mathcal{N}_i}\big)}{\partial \mathbf{z}_j}, \tag{7}$$

*where the partial derivatives $\frac{\partial f}{\partial \mathbf{z}} = [\frac{\partial f_i}{\partial \mathbf{z}_j}]_{ij}$ are sparse* (see Appendix A.1 for the derivation).

### 3.3 Equivariant local differential

Our goal is to have a differential function $f$ that is a PDE operator used in Equation 4, and that satisfies the natural equivariances and invariances of the molecules. Specifically, this function must be (i) translation equivariant: translating the input results in an equivalent translation of the output; (ii) rotational (and reflection) equivariant: rotating the input results in an equivalent rotation of the output; and (iii) permutation equivariant: permuting the input results in the same permutation of the output. Therefore, we chose to use E(3)-Equivariant GNN (EGNN) (Satorras et al., 2021), which is translation, rotation and reflection equivariant (E(n)), and permutation equivariant with respect to an input set of points (see Appendix A.3 for details). EGNN takes as input the node embeddings as well as the geometric information (polar coordinates (2D) and spherical polar coordinates (3D)). Interestingly, `ModFlow` can be viewed as a message passing temporal graph network (Rossi et al., 2020; Souza et al., 2022) as shown next.

**Proposition 2:** *Modular Flows can be cast as message passing Temporal Graph Networks (TGNs). The operations are listed in Table 2, where `ModFlow` is subjected to a single layer of EGNN. (See Appendix A.4 for more details).*

Table 2: `ModFlow` as a temporal graph network (TGN). Adopting notation for TGNs from Rossi et al. (2020) $v_i$ is a node-wise event on $i$; $e_{ij}$ denotes an (asymmetric) interaction between $i$ and $j$; $\mathbf{s}_i$ is the memory of node $i$; and $t$ and $t^-$ denote time with $t^-$ being the time of last interaction before $t$, e.g., $\mathbf{s}_i(t^-)$ is the memory of $i$ just before time $t$; and msg and agg are learnable functions (e.g., MLP) to compute, respectively, the individual and the aggregate messages. For `ModFlow`, we use $\mathbf{r}_{ij}$ to denote the spatial distance $\mathbf{x}_i - \mathbf{x}_j$, and $a_{ij}$ to denote the attributes of the edge between $i$ and $j$. The functions $\phi_e$, $\phi_x$, and $\phi_h$ are as defined in (Satorras et al., 2021).

| Method | TGN layer | ModFlow |
|---|---|---|
| Edge | $\mathbf{m}'_{ij}(t) = \operatorname{msg}\left(\mathbf{s}_i(t^-), \mathbf{s}_j(t^-), \Delta t, \mathbf{e}_{ij}(t)\right)$ 
 $\overline{\mathbf{m}}'_i(t) = \operatorname{agg}\left(\{\mathbf{m}'_{ij}(t) \mid j \in \mathcal{N}_i\}\right)$ | $\mathbf{m}_{ij}(t) = \phi_e\left(\mathbf{z}_i(t), \mathbf{z}_j(t), \|\mathbf{r}_{ij}(t)\|^2, a_{ij}\right)$ 
 $\mathbf{m}_i(t) = \sum_{j \in \mathcal{N}(i)} \mathbf{m}_{ij}$ 
 $\hat{\mathbf{m}}_{ij}(t) = \mathbf{r}_{ij}(t) \cdot \phi_x\left(\mathbf{m}_{ij}(t)\right)$ 
 $\hat{\mathbf{m}}_i(t) = C \sum_{j \in \mathcal{N}(i)} \hat{\mathbf{m}}_{ij}(t)$ |
| Memory state | $\mathbf{s}_i(t) = \operatorname{mem}\left(\overline{\mathbf{m}}'_i(t), \mathbf{s}_i(t^-)\right)$ | $\mathbf{x}_i(t+1) = \mathbf{x}_i(t) + \hat{\mathbf{m}}_i(t)$ |
| Node | $\mathbf{z}'_i(t) = \sum_{j \in \mathcal{N}_i} h\left(\mathbf{s}_i(t), \mathbf{s}_j(t), \mathbf{e}_{ij}(t), \mathbf{v}_i(t), \mathbf{v}_j(t)\right)$ | $\mathbf{z}_i(t+1) = \phi_h\left(\mathbf{z}_i(t), \mathbf{m}_i(t)\right)$ |

## 3.4 Training objective

Normalizing flows are predominantly trained to minimize $\mathrm{KL}[p_{\mathrm{data}}||p_\theta]$, i.e., the KL-divergence between the unknown data distribution $p_{\mathrm{data}}$ and the flow-generated distribution $p_\theta$. This objective is equivalent to maximizing $\mathbb{E}_{p_{\mathrm{data}}}[\log p_\theta]$ (Papamakarios et al., 2021). However, note that the discrete graphs $G$ and the continuous atom scores $\mathbf{z}(t)$ reside in different spaces. Thus, in order to apply flows, a mapping between the observation space and the flow space is needed. Earlier approaches use dequantisation to turn a graph $G$ into a distribution of latent states, and argmax to deterministically map latent states to graphs (Zang and Wang, 2020).

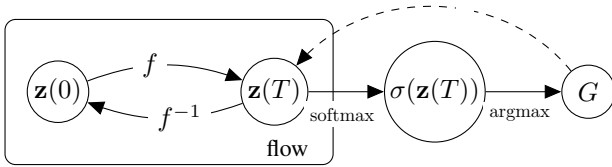

Figure 3: Plate diagram showing both the inference and generative components of `ModFlow`.

We instead reduce the learning problem to maximizing $\mathbb{E}_{\hat{p}_{\mathrm{data}}(\mathbf{z}(T))}[\log p_\theta(\mathbf{z}(T))]$, where we turn the observed set of graphs $\{G_n\}$ into a set of scores $\{\mathbf{z}_n\}$ using

$$\mathbf{z}_n(G_n; \epsilon) = (1 - \epsilon)\,\mathrm{onehot}(G_n) \;+\; \frac{\epsilon}{|\mathcal{A}_f|}\mathbf{1}_{M(n)}\mathbf{1}_{|\mathcal{A}_f|}^\top \,,$$

where $\mathrm{onehot}(G_n)$ is a matrix of size $M(n) \times |\mathcal{A}_f|$ (i.e., rows equal to the number of nodes in $G_n$ and columns equal to the number of possible node labels) such that $G_n(i, k) = 1$ if $v_i = a_k \in \mathcal{A}_f$, that is if the vertex $i$ is labeled with atom $k$, and 0 otherwise; $\mathbf{1}_q$ is a vector with $q$ entries each set to 1; $\mathcal{A}_f \in \{\mathcal{A}, \mathcal{A}_{\mathrm{tree}}\}$; and $\epsilon \in [0, 1]$ is added to model the noise in estimating the posterior $p(\mathbf{z}(T)|G)$ due to short-circuiting the inference process from $G$ to $\mathbf{z}(T)$ skipping the intermediate dependencies, thereby inducing an unconditional distribution $\hat{p}_{\mathrm{data}}$ that is slightly different from the true data distribution $p_{\mathrm{data}}$. The plate diagram in Figure 3 summarizes the overall procedure.

Effectively, we exploit the (non-reversible) composition of the argmax and softmax operations to transition from the continuous flow space to the discrete graph space, but skip this composition altogether in the reverse direction. Importantly, this short-circuiting allows `ModFlow` to keep the forward and backward flows between $\mathbf{z}(0)$ and $\mathbf{z}(T)$ completely aligned (i.e., reversible) unlike previous approaches. We maximize the following objective over $N$ training graphs:

$$\underset{\theta}{\mathrm{argmax}}\,\mathcal{L} = \mathbb{E}_{\hat{p}_{\mathrm{data}}(\mathbf{z})}\log p_\theta(\mathbf{z}) \tag{8}$$

$$\approx \frac{1}{N}\sum_{n=1}^{N}\log p_T\big(\mathbf{z}(T) = \mathbf{z}_n\big) \tag{9}$$

$$= \frac{1}{N}\sum_{n=1}^{N}\left(\sum_{i=1}^{M(n)}\log p_0(\mathbf{z}_i(0)) - \sum_{i=1}^{M(n)}\int_0^T \mathrm{tr}\frac{\partial f_\theta(t, \mathbf{z}_i(t), \mathbf{z}_{\mathcal{N}_i}(t), \mathbf{x}_i, \mathbf{x}_{\mathcal{N}_i})}{\partial \mathbf{z}_i(t)}dt\right)\,, \tag{10}$$

which factorizes over the size $M(n)$ of the $n$'th training molecule. The encoding probability follows from Equation 6, where $\mathbf{z}(0)$ can be traced by traversing the flow $f$ backward in time starting from $\mathbf{z}_n$ at time $t = T$ until $t = 0$. In practice we solve ODE integrals using a numerical solver such as Runge-Kutta. We thus delegate this task to a general solver of the form `ODESolve`$(\mathbf{z}, f_\theta, T)$, where map $f_\theta$ is applied for $T$ steps starting with $\mathbf{z}$. An optimizer `optim` is also required for updating $\theta$.

## 3.5 Molecular generation

Given a molecular structure, we can generate novel molecules by sampling an initial state $\mathbf{z}(0) \sim \mathcal{N}(0, I)$, and running the modular flow forward in time for $T$ steps and obtain $\mathbf{z}(T)$. This procedure maps a tractable base distribution $p_0$ to a more complex distribution $p_T$. We follow argmax to pick the most probable label assignment for each node (Zang and Wang, 2020). We outline the procedures for training and generation in Algorithm 1 and Algorithm 2 respectively.

**Algorithm 1** Training `ModFlow`

**Require:** Dataset $\mathcal{D}$, iterations $n_{\text{iter}}$, batch size $B$, number of batches $n_B$
1: Initialise parameters $\theta$ of `ModFlow` (EGNN)
2: **for** $k = 1, \ldots, n_{\text{iter}}$ **do**
3:     **for** $b = 1, \ldots, n_B$ **do**
4:         Sample $\mathcal{D}_b = \{G_1, \ldots, G_B\}$ from $\mathcal{D}$
5:         Define $\mathbf{z}_b(T) := \{\mathbf{z}_r(T) : G_r \in \mathcal{D}_b\}$
6:         Set $\mathbf{z}_b(T)$ to $\mathbf{z}_b(G_b; \epsilon)$
7:         $\mathcal{L}_b = \dfrac{1}{B} \sum\limits_{G_r \in D_b} \log p_\theta(\mathbf{z}_r(T)), \text{ using}$

        $\mathbf{z}_b(0) = \texttt{ODESolve}(\mathbf{z}_b(T), f_\theta^{-1}, T)$
8:     **end for**
9:     $\theta \leftarrow \texttt{optim}(\frac{1}{n_B} \sum_{b=1}^{n_B} \mathcal{L}_b; \theta)$
10: **end for**

**Algorithm 2** Generating with `ModFlow`

1: Sample $\mathbf{z}(0) \sim \mathcal{N}(0, I)$
2: $\mathbf{z}(T) = \texttt{ODESolve}(\mathbf{z}(0), f_\theta, T)$
3: Assign labels by $\text{argmax}(\sigma(\mathbf{z}(T)))$

## 4 Experiments

We first demonstrate the ability of Modular Flows (`ModFlow`) to learn highly discontinuous synthetic patterns on 2D grids. We also evaluated `ModFlow` models trained, variously, on (i) 2D coordinates, (ii) 3D coordinates, (iii) 2D coordinates + tree representation, and (iv) 3D coordinates + tree representation on the tasks of molecular generation and optimization. Our results show that `ModFlow` compares favorably to other prominent flow and non-flow based molecular generative models, including GraphDF (Luo et al., 2021), GraphNVP (Madhawa et al., 2019), MRNN(Popova et al., 2019), and GraphAF (Shi et al., 2020). Notably, `ModFlow` achieves state-of-the-art results without validity checks or post hoc correction. We also provide results of our ablation studies to underscore the relevance of geometric features and equivariance toward this superlative empirical performance.

### 4.1 Density Estimation

We generated our synthetic data in the following way. We considered two variants of a chessboard pattern, namely, (i) $4 \times 4$ grid where every node takes a binary value, 0 or 1, and neighboring nodes have different values; and (ii) $16 \times 16$ grid where nodes in each block of $4 \times 4$ all take the same value (0 or 1), different from the adjacent blocks. We also experimented with a $20 \times 20$ grid describing alternating stripes of 0s and 1s.

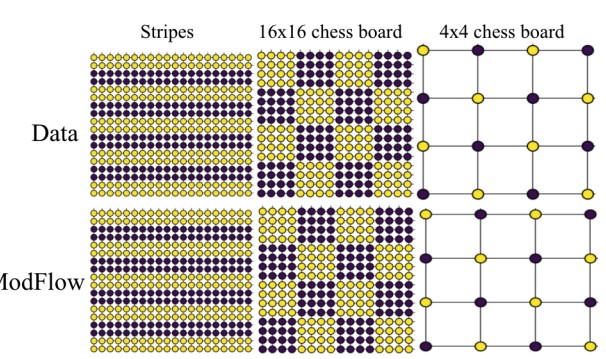

Figure 4: `ModFlow` can accurately learn to reproduce complex, discontinuous graph patterns.

Figure 4 shows that `ModFlow` can learn neural differential functions $f_\theta$ that reproduce the patterns almost perfectly, indicating sufficient capacity to model complex patterns. That is, `ModFlow` is able to transform the initial Gaussian distribution into different multi-modal and discontinuous distributions.

### 4.2 Molecule Generation

**Data.** We trained and evaluated all the models on ZINC250k (Irwin et al., 2012) and QM9 (Ramakrishnan et al., 2014) datasets. The ZINC250k set contains 250,000 drug-like molecules, each consisting of up to 38 atoms. The QM9 set contains 134,000 stable small organic molecules with atoms from the set $\{C, H, O, N, F\}$. The molecules are processed to be in the *kekulized* form with hydrogens removed by the RDkit software (Landrum et al., 2013).

Table 3: Random generation on QM9 (top) and ZINC250K (bottom) without post hoc validity corrections. Results with * are taken from Luo et al. (2021). Higher values are better for all columns.

| Method | Validity % | Uniqueness % | Novelty % | Reconstruction % |
|---|---|---|---|---|
| GVAE | 60.2 | 9.3 | 80.9 | 96.0 |
| GraphNVP* | 83.1 | 99.2 | 58.2 | 100 |
| GRF* | 84.5 | 66 | 58.6 | 100 |
| GraphAF* | 67 | 94.2 | 88.8 | 100 |
| GraphDF* | 82.7 | 97.6 | 98.1 | 100 |
| MoFlow* | 89.0 | 98.5 | 96.4 | 100 |
| ModFlow (2D-EGNN) | **96.2** $\pm 1.7$ | **99.5** | **100** | 100 |
| ModFlow (3D-EGNN) | **98.3** $\pm 0.7$ | 99.1 | **100** | 100 |
| ModFlow (JT-2D-EGNN) | **97.9** $\pm 1.2$ | 99.2 | **100** | 100 |
| ModFlow (JT-3D-EGNN) | **99.1** $\pm 0.8$ | 99.3 | **100** | 100 |

| Method | Validity % | Uniqueness % | Novelty % | Reconstruction % |
|---|---|---|---|---|
| MRNN | 65 | 99.89 | 100 | n/a |
| GVAE | 7.2 | 9 | 100 | 53.7 |
| GCPN* | 20 | 99.97 | 100 | n/a |
| GraphNVP* | 42.6 | 94.8 | 100 | 100 |
| GRF* | 73.4 | 53.7 | 100 | 100 |
| GraphAF* | 68 | 99.1 | 100 | 100 |
| GraphDF* | 89 | 99.2 | 100 | 100 |
| MoFlow* | 50.3 | **99.9** | 100 | 100 |
| ModFlow (2D-EGNN) | **94.8** $\pm 1.0$ | 99.4 | 100 | 100 |
| ModFlow (3D-EGNN) | **95.4** $\pm 1.2$ | 99.7 | 100 | 100 |
| ModFlow (JT-2D-EGNN) | **97.4** $\pm 1.4$ | 99.1 | 100 | 100 |
| ModFlow (JT-3D-EGNN) | **98.1** $\pm 0.9$ | 99.3 | 100 | 100 |

**Setup.** We adopt common quality metrics to evaluate molecular generation. *Validity* is the fraction of molecules that satisfy the respective chemical valency of each atom. *Uniqueness* refers to the fraction of generated molecules that is unique (i.e, not a duplicate of some other generated molecule). *Novelty* is the fraction of generated molecules that is not present in the training data. *Reconstruction* is the fraction of molecules that can be reconstructed from their encoding. Here, we strictly limit ourselves to comparing all methods on their validity scores *without resorting to external correction*. We trained each model with 5 random weight initializations, and generated 50,000 molecular graphs for evaluation. We report the mean and the standard deviation scores across these multiple runs.

**Implementation.** The models were implemented in PyTorch (Paszke et al., 2019). The EGNN method used only a single layer with embedding dimension of 32. We trained with the Adam optimizer (Kingma and Ba, 2014) for 50-100 epochs (until the training loss became stable), with batch size 1000 and learning rate 0.001. ModFlow is significantly faster compared to autoregressive models such as GraphAF and GraphDF. For more details, see Appendix A.5.

**Results.** Table 3 reports the performance on QM9 (top) and ZINC250K (bottom) respectively. ModFlow achieves state-of-the-art results across all metrics. Notably, its reconstruction rate is 100% (similar to other flow models); in addition, however, the novelty (100%) and uniqueness scores ($\approx$99%) are also very high. Moreover, ModFlow surpassed the other methods on validity (95%-99%).

In Appendix A.6, we document additional evaluations with respect to the MOSES metrics that access the overall quality of generated molecules, as well as the distributions of chemical properties. All these results substantiate the promise of ModFlow as an effective tool for molecular generation.

## 4.3 Property-targeted Molecular Optimization

The task of molecular optimization is to search for molecules that have better chemical properties. We choose the standard quantitative estimate of drug-likeness (QED) as our target chemical property. QED measures the potential of a molecule to be characterized as a drug. We used a pre-trained

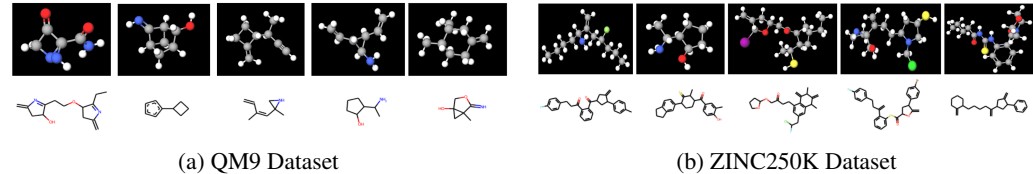

(a) QM9 Dataset            (b) ZINC250K Dataset

Figure 5: Samples of molecules generated by `ModFlow`. More examples are shown in Appendix A.7.

ModFlow model $f$ to encode molecules $\mathcal{M}$ into their embeddings $\mathcal{Z} = f(\mathcal{M})$, and applied linear regression to obtain QED scores $\mathcal{Y}$ from these embeddings. We then interpolate in the latent space of each molecule along the direction of increasing QED via several gradient ascent steps, i.e., updates of the form $\mathcal{Z}' = \mathcal{Z} + \lambda * \frac{d\mathcal{Y}}{d\mathcal{Z}}$, where $\lambda$ denotes the length of the search step. The final embedding thus obtained is decoded as a new molecule via the reverse mapping $f^{-1}$.

QED: 0.332        QED: 0.428        QED: 0.557

Figure 6: Example of chemical property optimization on the ZINC250K dataset. Given the left-most molecule, we interpolate in latent space along the direction which maximizes its QED property.

QED: 0.355     QED: 0.419     QED: 0.552     QED: 0.614

Figure 7: Example of chemical property optimization on the QM9 dataset. Given the left-most molecule, we interpolate in latent space along the direction which maximizes its QED property.

Table 4: Performance in terms of the best QED scores (baselines are taken from Luo et al. (2021)).

| Method | 1st | 2nd | 3rd |
|---|---|---|---|
| ZINC (dataset) | 0.948 | 0.948 | 0.948 |
| JTVAE | 0.925 | 0.911 | 0.910 |
| GCPN | 0.948 | 0.947 | 0.945 |
| MRNN | 0.844 | 0.799 | 0.736 |
| GraphAF | 0.948 | 0.948 | 0.947 |
| GraphDF | 0.948 | 0.948 | 0.948 |
| MoFlow | 0.948 | 0.948 | 0.948 |
| ModFlow (2D-EGNN) | 0.948 | 0.941 | 0.937 |
| ModFlow (3D-EGNN) | 0.948 | 0.937 | 0.931 |
| ModFlow (JT-2D-EGNN) | 0.947 | 0.941 | 0.939 |
| ModFlow (JT-3D-EGNN) | 0.948 | 0.948 | 0.945 |

Figure 6 and Figure 7 show examples of the molecules decoded from the learned latent space using this procedure, starting with molecules having a low QED score. Note that the number of valid molecules decoded back varies on the query molecule. We report the discovered novel molecules sorted by their QED scores in Table 4. Clearly, `ModFlow` is able to find novel molecules with high QED scores.

## 4.4 Ablation Studies

We also performed ablation experiments to gain further insights about `ModFlow`, as we describe next.

**E(3)-equivariant versus not equivariant.** Molecules exhibit translational and rotational symmetries, so we conducted an ablation study to quantify the effect of incorporating these symmetries in our model. We compare the results obtained using an EGNN with a non-equivariant graph convolutional network (GCN). For our purpose, we used a 3-layer GCN with layer sizes 64-32-32. The validity scores in Table 5 provide strong evidence in favor of modeling the symmetries explicitly in the proposed Modular Flows.

Table 5: Random generation performance on ZINC250K and QM9 dataset with E(3)-EGNN vs GCN.

| Dataset | Method | Validity % | Uniqueness % | Novelty % |
|---|---|---|---|---|
| ZINC250K | `ModFlow` (3D-EGNN) | $95.4 \pm 1.2$ | 99.7 | 100 |
| | `ModFlow` (GCN) | $90.3 \pm 1.9$ | 99.7 | 100 |
| QM9 | `ModFlow` (3D-EGNN) | $98.3 \pm 0.7$ | 99.1 | 100 |
| | `ModFlow` (GCN) | $93.3 \pm 0.5$ | 98.8 | 100 |

**2D versus 3D.** Finally, we study whether including information about the 3D coordinates improves the model. Note that the EGNN-coupled differential function obtains either the 2D or 3D positions as polar coordinates, where the 3D positions have an extra degree of freedom. Table 6 shows that transitioning from 2D to 3D improves the mean validity score.

Table 6: Random generation on ZINC250K and QM9 dataset with 2D versus 3D features.

| Dataset | Method | Validity % | Uniqueness % | Novelty % |
|---|---|---|---|---|
| ZINC250K | `ModFlow` (3D-EGNN) | $95.4 \pm 1.2$ | 99.7 | 100 |
| | `ModFlow` (2D-EGNN) | $94.8 \pm 1.0$ | 99.4 | 100 |
| QM9 | `ModFlow` (3D-EGNN) | $98.3 \pm 0.7$ | 99.1 | 100 |
| | `ModFlow` (2D-EGNN) | $96.2 \pm 1.7$ | 99.5 | 100 |

## 5 Conclusion

We proposed `ModFlow`, a new generative flow model where multiple flows interact locally according to a coupled ODE, resulting in accurate modeling of graph densities and high quality molecular generation without any validity checks or correction. Interesting avenues open up, including the design of (a) more nuanced mappings between discrete and continuous spaces, and (b) extensions of modular flows to (semi-)supervised settings.

## 6 Acknowledgments

The calculations were performed using resources within the Aalto University Science-IT project. This work has been supported by the Academy of Finland under the *HEALED* project (grant 13342077).

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
