# OpenReview forum: "Modular Flows: Differential Molecular Generation"
_NeurIPS.cc/2022/Conference — NeurIPS 2022 Accept_

### Official Review · Reviewer_HLkc · 2022-07-11

**Rating:** 6
**Confidence:** 3
**Soundness:** 3 good
**Presentation:** 3 good
**Contribution:** 3 good

**Summary:**

This work introduces ModFlow, a modular continuous normalizing flow model parametrized by an E(3) equivariant GNN. To demonstrate the efficacy of ModFlow, the model is trained on QM9 and ZINC250K to the end of generating valid, unique, and novel samples.

**Questions:**

1. Line 156, 157: should these be rotational equivariant and permutation equivariant instead of invariant?
2. Line 180-183: what does it mean for the forward and backward flows to be completely aligned?
3. To the best of my knowledge, the ZINC250k dataset only contains 2D atom coordinates. Are the generated 3D structures shown in Figure 5 ZINC250K computed via additional libraries, or are they generated by ModFlow? Similarly, are the QM9 conformers generated by ModFlow, or are they computed based on the molecular graph?
4. How does ModFlow compare to other models in terms of training/inference time? How does the training/inference time of ModFlow scale with molecule size (QM9 \to ZINC250K)?
5. I understand that it is appealing to have a model that captures chemical composition laws from data rather than relying on e.g. valency checks from external libraries. However, I am wondering if there is an example application in which a model's intrinsic generative validity matters?

**Limitations:**

While ModFlow can generate molecules with high validity by directly sampling from the latent space, there does not seem to be a straightforward way to incorporate valency checks into the sampling procedure of ModFlow, should one desire to do so.

**Strengths And Weaknesses:**

Strengths:
1. The writing is clear for the most part, and the model is well-presented.
2. The validity, uniqueness, and novelty results for QM9 and ZINC250K look great. The generated samples also seem reasonable.

Weaknesses:
My main concerns with this work are related to its numerical experiments.
1. More metrics should be included for a comprehensive assessment of ModFlow's generation quality. This is because validity, uniqueness, and novelty scores may not correlate with how realistic the samples are. To this end, one may use the Frechet ChemNet Distance (FCD), which measures the similarity between the set of generated molecules and the set of training molecules. One can easily compute the FCD as well as a variety of other metrics through MOSES: https://github.com/molecularsets/moses, and the inclusion of these additional metrics will make the paper much more convincing.
2. More experiments are needed to fully demonstrate the effectiveness of ModFlow. For example, property optimization is conducted in e.g. JTVAE and GraphAF to find the best generated molecules with certain chemical scores, e.g. penalized LogP, QED. Such optimization showcases that the model is able to learn a meaningful latent space useful for downstream tasks. Other possibilities are latent space interpolation and property prediction.

Edit: in light of the added experimental results during the rebuttal, I have now raised my score to 6 as I believe the paper makes a solid contribution in cheminformatics.

---

> ### Author Response · Authors · 2022-08-02
> **Reply to comments and suggestions**
>
> Thank you very much for many excellent suggestions! We have acted on all of these, and additionally address all your questions and comments below.
>
> > More metrics should be included for a comprehensive assessment of ModFlow's generation quality. This is because validity, uniqueness, and novelty scores may not correlate with how realistic the samples are. To this end, one may use the Frechet ChemNet Distance (FCD), which measures the similarity between the set of generated molecules and the set of training molecules. One can easily compute the FCD as well as a variety of other metrics through MOSES: https://github.com/molecularsets/moses, and the inclusion of these additional metrics will make the paper much more convincing
>
> **Evaluation with MOSES metrics**: We agree that more metrics should be included for a comprehensive assessment of ModFlow. Based on your feedback, we have included four different MOSES metrics, and report performance with respect to these.
> - **Frechet Chemnet distance (FCD)**: a general purpose metric that measures diversity as well as chemical and biological property alignment between generated and reference molecules. _Lower is better_
> - **Frag**: looks at BRICS fragment matching between generated and reference molecules. _Higher is better_
> - **Similarity to nearest neighbor (SSN)**: Looks at closest distance between generated and reference molecules, and quantifies how close the generated molecules are to the true molecule manifold. _Higher is better_
> - **Internal diversity (IntDiv)**: accounts for diversity of the generated set. _Higher is better_
>
> We computed the above metrics for the proposed method (ModFlow) and 5 state-of-the-art methods including both flow-based (GraphAF, GraphDF, MoFlow) and non-flow (GVAE, JTVAE, GraphEBM) methods. **ModFlow achieves best results across these metrics on both the ZINC250K and QM9 datasets** as shown in the tables below. For instance, on QM9, the scores are $\approx$ (0.40 vs 0.50 for best competing) on FCD, (0.93 vs 0.89 best competing) on Frag, (0.62 vs 0.58 best competing) on SSN, and (0.88 vs 0.85 best competing) on IntDiv.  Notably, ModFlow registers a lower FCD score and higher IntDiv value compared to other methods, which suggests it is able to generate diverse set of molecules similar to those present in dataset. We will include a discussion about the new metrics, and add these results as a table in the final version.
>
> For QM9
> |Method|FCD|Frag|SNN|IntDiv|
> |-----|----|-----|-----|------|
> |GVAE|0.513|0.821|0.582|0.822|
> |GraphEBM  |0.551|0.831|0.547|0.831|
> |GraphAF|0.732 | 0.863| 0.565 | 0.823|
> |GraphDF|0.683 |0.892|0.562 |0.839|
> |MoFlow|0.496 | 0.840 |0.502 |  0.852|
> |ModFlow(2D-EGNN)|0.432| 0.928 | 0.608 | 0.875|
> |ModFlow(3D-EGNN)|0.478| 0.934 | 0.613 | 0.885|
> |ModFlow(JT-2D-EGNN)| 0.421| 0.921| 0.595 | 0.867|
> |ModFlow(JT-3D-EGNN)|**0.401** | **0.939** | **0.624** | **0.889**
>
> For ZINC:
> |Method|FCD|Frag|SNN|IntDiv|
> |-----|----|-----|-----|------|
> |JTVAE|0.512|0.890|0.5477|0.855|
> |GVAE|0.571|0.871|0.532|0.852|
> |GraphEBM|0.613|0.843|0.487|0.821|
> |GraphAF|0.524|0.803|0.465|0.855|
> |GraphDF|0.658|0.869|0.515|0.829|
> |MoFlow|0.597|0.851|0.452|0.832|
> |ModFlow(2D-EGNN)|**0.495** | 0.891 | 0.570 | 0.863|
> |ModFlow(3D-EGNN)|0.512| 0.905 | 0.584 | 0.869|
> |ModFlow(JT-2D-EGNN)| 0.501| 0.915 | 0.563 | 0.857|
> |ModFlow(JT-3D-EGNN)|0.523| **0.929** | **0.594** | **0.879**|
>
> > I understand that it is appealing to have a model that captures chemical composition laws from data rather than relying on e.g. valency checks from external libraries. However, I am wondering if there is an example application in which a model's intrinsic generative validity matters?
>
> **Why intrinsic generative validity matters?**
>
> Thank you for an important question! We care about intrinsic generative validity due to several reasons:
> - In the practical drug-discovery pipeline, for the task of {\em de novo} drug design, any generated non-valid molecule has to be discarded, which reduces the sampling efficiency of the procedure. If the model is not able to account for the underlying valid molecule manifold, it likely has not yet learned the intrinsic structure of molecules properly.
> - Relying on valence checks and post hoc corrections when intrinsic validity is low leads to a considerable distributional shift between the training and generated distributions, which in turn restricts the applicability of the methods from a practical perspective.
> - If the model does not understand when a molecule is valid or invalid, it likely also struggles understanding higher level concepts, such as factors responsible for molecular properties. These issues often translate into poor performance on downstream tasks (such as property prediction, eg. QSAR).

---

> > ### Author Response · Authors · 2022-08-02
> > **Reply to comments and suggestions**
> >
> > >More experiments are needed to fully demonstrate the effectiveness of ModFlow. For example, property optimization is conducted in e.g. JTVAE and GraphAF to find the best generated molecules with certain chemical scores, e.g. penalized LogP, QED. Such optimization showcases that the model is able to learn a meaningful latent space useful for downstream tasks. Other possibilities are latent space interpolation and property prediction.
> >
> > **Results with property optimization**: Thank you for another meaningful set of experiments. Based on your comments, we conducted  property-targeted molecular optimization. As we describe below, the results further demonstrate the efficacy of ModFlow.
> >
> > We begin with the setup. As you mentioned, the central objective of property optimization is to search for molecules with a better chemical score, starting from a similar(in the latent space) but inferior molecule with respect to the chemical property. We used QED score as our target chemical property. To discover better molecules, we first trained a linear regression model to map molecule latent embeddings to the corresponding QED score.
> >
> > We selected some molecules with low QED scores and search in the vicinity of their latent representations along the direction of increasing QED, before finally decoding the latent embedding back to a new molecule. The results are shown for the ZINC250K data in the following table. Notably, ModFlow is able to find novel molecules with high QED scores. We also report the discovered novel molecules - sorted by their QED scores - in section A.7 in the Supplementary of the current version.
> >
> > Property optimization performance evaluated by best QED scores.
> > |Method| 1st | 2nd | 3rd |
> > |---------|-----|----|----|
> > |ZINC (dataset)  | 0.948  |0.948  | 0.948|
> > |JTVAE| 0.925 | 0.911| 0.910|
> > |GCPN | 0.948 | 0.947| 0.945|
> > |MRNN |0.844 | 0.799| 0.736|
> > |GraphAF | 0.948 | 0.948| 0.947|
> > |GraphDF  | 0.948 | 0.948| 0.948|
> > |MoFlow | 0.948  |0.948|0.948|
> > |ModFlow (2D-EGNN)  |0.948 | 0.941 | 0.937|
> > |ModFlow (3D-EGNN)  |0.948 | 0.937 | 0.931|
> > |ModFlow (JT-2D-EGNN)  |0.947 |0.941 | 0.939|
> > |ModFlow (JT-3D-EGNN)  |0.948 |0.948 |0.945|
> >
> > >To the best of my knowledge, the ZINC250k dataset only contains 2D atom coordinates. Are the generated 3D structures shown in Figure 5 ZINC250K computed via additional libraries, or are they generated by ModFlow? Similarly, are the QM9 conformers generated by ModFlow, or are they computed based on the molecular graph?
> >
> > We computed the 3D coordinates for both QM9 and ZINC250k with the RDKit library.
> >
> > > How does ModFlow compare to other models in terms of training/inference time? How does the training/inference time of ModFlow scale with molecule size (QM9 $\to$ ZINC250K)?
> >
> > **Training and inference time comparisons**:  We present here a comparison of the runtimes of different models. Notably, the proposed model, ModFlow, turns out to be fastest in terms of the sampling time, with around 0.50 seconds per generated molecule (on ZINC250k model). The MoFlow, GraphAF, GVAE and GraphEBM are all around 50$\%$ slower, while GraphDF is around 5x slower. Similar results hold for the QM9 model. A major reason for the computational efficiency is the neighborhood locality of our model, and the low-dimensionality of the latent variables (only 9 latent scalars per node). The training time varies with the implementation, training procedure and computing resources. The proposed model, ModFlow, took approximately 1.5 - 2 hours for one epoch to train on a Tesla V100 for 250,000 molecules. We trained ModFLow for 50 - 100 epochs. The training time of our model is only slightly more than some one-shot discrete models that use a single flow  unlike the the current proposal that associates an ODE flow with each node.
> > Notably, the training time is less than as compared to the autoregressive methods. We will add a detailed discussion on computational time requirements of the different methods in the revised version.
> >
> > Generation time of various models performed on QM9 and ZINC250K dataset. Time is in seconds denoted as $\mathcal{O}$(second/molecule)
> >
> > |Method |ZINC250K | QM9|
> > |-------|----------|------|
> > |GraphEBM |1.12 $\pm$ 0.34 |0.53 $\pm$ 0.16|
> > |GVAE      | 0.86 $\pm$ 0.12  |0.46 $\pm$ 0.07|
> > |GraphAF  |0.93 $\pm$ 0.14 | 0.56 $\pm$ 0.12 |
> > |GraphDF  |3.12 $\pm$ 0.56 | 1.92 $\pm$ 0.42|
> > |MoFlow | 0.71 $\pm$ 0.14  |  0.31 $\pm$ 0.04|
> > |ModFlow (2D-EGNN)  |0.46 $\pm$ 0.09 | 0.16 $\pm$ 0.04|
> > |ModFlow (3D-EGNN)  |0.55 $\pm$ 0.13 | 0.24 $\pm$ 0.06|
> > |ModFlow (JT-2D-EGNN)  |0.53 $\pm$ 0.07 | 0.21 $\pm$ 0.07|
> > |ModFlow (JT-3D-EGNN)  |0.62 $\pm$ 0.11 & 0.28 $\pm$ 0.09|
> >
> > > Line 156, 157: should these be rotational equivariant and permutation equivariant instead of invariant?
> >
> > Thank you for your comment. We have updated that section to focus only on equivariances (that we care about in this work).

---

> > > ### Author Response · Authors · 2022-08-02
> > > **Reply to comments and suggestions**
> > >
> > > > While ModFlow can generate molecules with high validity by directly sampling from the latent space, there does not seem to be a straightforward way to incorporate valency checks into the sampling procedure of ModFlow, should one desire to do so
> > >
> > > **Incorporating valency checks.**
> > > Yes, that's correct. The latent variables represent “partial” atoms or substructures, so it is difficult to incorporate validity checks during the evolution of flow directly.
> > >
> > > Still, in principle, one could incorporate validity checks once the molecules are generated using an extra loss, and backpropagate this information to the generative parameters despite the challenges arising due to validity being effectively a binary check.
> > >
> > > That said, we would like to emphasize that a main goal of this work has been to design a highly accurate  generator that obviates the need for any validity checks altogether!
> > >
> > > >Line 180-183: what does it mean for the forward and backward flows to be completely aligned?
> > >
> > > Normalizing flows are based on the invertibility of a sequence of transformations from a base distribution to a complex one. Note that this invertibility of transformations ensure that the is preserved along the flow in either direction (unlike, e.g., other likelihood based methods such as Variational Autoencoders). We refer to this forward/backward compatibility as alignment here.
> > >
> > > Thank you, again, for your constructive feedback! We hope we've sufficiently addressed your concerns, and if so, the same would translate into a raised score.

---

> > > > ### Comment · Reviewer_HLkc · 2022-08-07
> > > > **Response to author rebuttals**
> > > >
> > > > Thank you for the detailed response. Overall, my concerns are addressed adaquately for the most part - the experiments are comprehensive, and the results look strong. In an ideal world, a third molecular experiment would also be nice, some suggestions are property prediction with latent representations and constrained property optimization (see e.g. GraphAF). In terms of novelty, I still find the works slightly limited as it is mostly a NODE parameterized by EGNN, and both are existent models. All things considered, I have raised my score to 6 as I think this paper now makes a solid contribution in cheminformatics, and I have enough confidence that the model can produce good results in other molecule-based experiments.

---

> > > > > ### Author Response · Authors · 2022-08-08
> > > > > **Reply to the reviewer comment**
> > > > >
> > > > > Many thanks for your engagement in the discussion, and additional suggestions (we’ll continue to work toward improving the paper for the final version). We appreciate your feedback, and are grateful for your support and for your confidence in the proposed method.

---

### Official Review · Reviewer_c5ye · 2022-07-12

**Rating:** 6
**Confidence:** 3
**Soundness:** 2 fair
**Presentation:** 3 good
**Contribution:** 3 good

**Summary:**

The authors propose a new generative model for molecules based on normalizing flows. The network is based on a set of ODEs (one per node), that are coupled together to form a PDE, which lets the model graph densities accurately. The model is also E(3) equivariant, which is an important physical property for molecules.

The paper presents experiments on QM9 and Zinc250K datasets and evaluate results on validity, uniqueness, novelty, and reconstruction metrics. The ModFlow method performs well on these metrics.

**Questions:**

1. Please include a discussion of the runtime of your model for training and inference.
2. How does this scale with the number of atoms? Can it effectively generate larger systems like materials or bio molecules?

**Limitations:**

Please discuss limitations in terms of computational efficiency, classes of atomic systems that can be generated by this method etc.
For example, this method cannot directly be used for materials where periodic boundary conditions need to be respected.

Also, potential negative societal impact (such as generating toxic materials, etc.) should be mentioned.

**Strengths And Weaknesses:**

Strengths
* As the ModFlow method is based on combining normalizing flows with neural ODEs, it can generate graphs in one-shot, and also provide density estimates (unlike GANs).
* The model is E(3) equivariant, which is a desirable property for molecules and other atomic systems.

Weaknesses
* It is not entirely clear how useful the metrics used in the paper are. As such they are weak metrics -- for example, validity simply measures the fraction of molecules that do not violate chemical valency rule. Most chemical applications require generating based on more complex properties, such as generating non-toxic drug molecules, generating synthesizable materials, etc. Therefore, it is unclear how useful this method is in practice.

---

> ### Author Response · Authors · 2022-08-02
> **Reply to comments and suggestions**
>
> Thanks so much for your thoughtful comments and excellent suggestions! We've acted on all of them as we describe below.
>
> > It is not entirely clear how useful the metrics used in the paper are. As such they are weak metrics -- for example, validity simply measures the fraction of molecules that do not violate the chemical valency rule. Most chemical applications require generating based on more complex properties, such as generating non-toxic drug molecules, generating synthesizable materials, etc. Therefore, it is unclear how useful this method is in practice.
>
> Thank you for the opportunity to clarify this, as well as for suggesting evaluation with additional metrics that we report shortly.
>
> **Significance of high intrinsic validity**: First, we would like to emphasize that despite validity being a seemingly easy metric that requires only valency adherence,  earlier works have consistently struggled to achieve good validity scores without resorting to valence checks and post hoc correction. Such corrections lead to a considerable distributional shift between the training and generated distributions, which in turn restricts the applicability of the methods from a practical perspective.  In fact, to our knowledge, the current work represents the first model that is capable of achieving up to ~99$\%$ intrinsic valence validity without any checks,  which demonstrates its ability to model the underlying densities accurately.
>
> **Evaluation with stronger metrics.**:  Validity by itself is however not a strong metric as you rightly pointed out from a practical perspective. We Therefore, based on your suggestion,  we have added the much stronger MOSES metrics described below:
>
>  - **Frechet Chemnet distance (FCD)**: a general purpose metric that measures diversity as well as chemical and biological property alignment between generated and reference molecules. _Lower is better_
> - **Frag**: looks at BRICS fragment matching between generated and reference molecules. _Higher is better_
> - **Similarity to nearest neighbor (SSN)**: looks at closest distance between generated and reference molecules, and quantifies how close the generated molecules are to the true molecule manifold. _Higher is better_
> - **Internal diversity (IntDiv)**: accounts for diversity of the generated set. _Higher is better_
>
> We computed the above metrics for the proposed method (ModFlow) and 5 state-of-the-art methods including both flow-based (GraphAF, GraphDF, MoFlow) and non-flow (GVAE, JTVAE, GraphEBM) methods. **ModFlow achieves best results across these metrics on both the ZINC250K and QM9 datasets** as shown in the tables below. For instance, on QM9, the scores are $\approx$ (0.40 vs 0.50 for best competing) on FCD, (0.93 vs 0.89 best competing) on Frag, (0.62 vs 0.58 best competing) on SSN, and (0.88 vs 0.85 best competing) on IntDiv.  Notably, ModFlow registers a lower FCD score and higher IntDiv value compared to other methods, which suggests it is able to generate diverse set of molecules similar to those present in dataset.
>
> We will include a discussion about the new metrics, and add these results as a table in the final version
> For QM9:
>
> |Method| FCD ($\downarrow$)| Frag ($\uparrow$)| SNN ($\uparrow$) | IntDiv ($\uparrow$)|
> |------|-------|-------|-------|-----|
> |GVAE   |  0.513    | 0.821    | 0.582    | 0.822|
> |GraphEBM | 0.551 | 0.831| 0.547 |  0.831|
> |GraphAF  | 0.732 | 0.863| 0.565 | 0.823 |
> |GraphDF  | 0.683 | 0.892|0.562  |  0.839|
> |MoFlow | 0.496  | 0.840 |0.502 |  0.852 |
> |ModFlow (2D-EGNN)  | 0.432 | 0.928 | 0.608 | 0.875|
> |ModFlow (3D-EGNN)  |0.478 | 0.934 | 0.613 | 0.885|
> |ModFlow (JT-2D-EGNN)  | 0.421| 0.921|0.595 | 0.867|
> |ModFlow (JT-3D-EGNN)  | **0.401** | **0.939** |**0.624**| **0.889**|
>
> For ZINC:
>
> |Method| FCD ($\downarrow$)| Frag ($\uparrow$)| SNN ($\uparrow$) | IntDiv ($\uparrow$)|
> |------|-------|-------|-------|-----|
> |JTVAE  |    0.512   |    0.890  |   0.5477  | 0.855|
> |GVAE   |  0.571   | 0.871    |  0.532     | 0.852|
> |GraphEBM | 0.613 | 0.843| 0.487 |  0.821|
> |GraphAF  | 0.524 | 0.803| 0.465 | 0.855|
> |GraphDF  | 0.658 | 0.869| 0.515 | 0.829|
> |MoFlow | 0.597  |0.851 |0.452 |0.832|
> |ModFlow (2D-EGNN)  | **0.495** | 0.891 | 0.570 | 0.863|
> |ModFlow (3D-EGNN)  |0.512| 0.905 | 0.584 | 0.869|
> |ModFlow (JT-2D-EGNN)  | 0.501| 0.915 | 0.563 | 0.857|
> |ModFlow (JT-3D-EGNN)  | 0.523| **0.929** |**0.594**| **0.879**|

---

> > ### Author Response · Authors · 2022-08-02
> > **Reply to comments and suggestions**
> >
> > >Please include a discussion of the runtime of your model for training and inference.
> >
> > **Runtime comparisons**: Thank you for another excellent suggestion! We have now conducted additional experiments to compare the runtimes of different models. Notably, the proposed model, ModFlow, turns out to be fastest in terms of the sampling time, with around 0.50 seconds per generated molecule (on ZINC250k model). The MoFlow, GraphAF, GVAE and GraphEBM are all around 50$\%$ slower, while GraphDF is around 5x slower. Similar results hold for the QM9 model. A major reason for the computational efficiency is the neighborhood locality of our model, and the low-dimensionality of the latent variables (only 9 latent scalars per node). The training time varies with the implementation, training procedure and computing resources. The proposed model, ModFlow, took approximately 1.5 - 2 hours for one epoch to train on a Tesla V100 for 250,000 molecules. We trained ModFLow for 50 - 100 epochs. The training time of our model is only slightly more than some one-shot discrete models that use a single flow
> > unlike the the current proposal that associates an ODE flow with each node.
> > Notably, the training time is less than as compared to the autoregressive methods. We will add a detailed discussion on computational time requirements of the different methods in the revised version.
> >
> >
> > Generation time of various models performed on QM9 and ZINC250K dataset. Time is in seconds denoted as $\mathcal{O}$(second/molecule):
> >
> > |Method |ZINC250K | QM9|
> > |-------|----------|------|
> > |GraphEBM |1.12 $\pm$ 0.34 |0.53 $\pm$ 0.16|
> > |GVAE      | 0.86 $\pm$ 0.12  |0.46 $\pm$ 0.07|
> > |GraphAF  |0.93 $\pm$ 0.14 | 0.56 $\pm$ 0.12 |
> > |GraphDF  |3.12 $\pm$ 0.56 | 1.92 $\pm$ 0.42|
> > |MoFlow | 0.71 $\pm$ 0.14  |  0.31 $\pm$ 0.04|
> > |ModFlow (2D-EGNN)  |0.46 $\pm$ 0.09 | 0.16 $\pm$ 0.04|
> > |ModFlow (3D-EGNN)  |0.55 $\pm$ 0.13 | 0.24 $\pm$ 0.06|
> > |ModFlow (JT-2D-EGNN)  |0.53 $\pm$ 0.07 | 0.21 $\pm$ 0.07|
> > |ModFlow (JT-3D-EGNN)  |0.62 $\pm$ 0.11 | 0.28 $\pm$ 0.09|
> >
> > > How does this scale with the number of atoms? Can it effectively generate larger systems like materials or bio molecules?
> >
> > **Scalability results**: Since ModFlow implements a continuous-time normalizing flow (CNF), the main computational bottleneck is the sequential nature of the forward process. Thus, the forward execution of an ODE requires a certain amount of time based on the number of update steps, akin to number of layers in a graph neural network, in the forward process, and the ODE solver (e.g. Runge-Kutta).
> >
> > However, running a larger coupled ODE with more atoms at once can be done in the same ``go”, with only linear or even faster scaling (until we hit the memory limits), akin to the parallelism benefits within each layer of a graph neural network. Thus, there are no inherent issues in scaling the model to much larger systems like the ones you mentioned. In particular, the local nature of the generative process should make this approach very amenable for such systems. Adapting ModFlow to such applications is an interesting and important research direction.
> >
> > >Please discuss limitations in terms of computational efficiency, classes of atomic systems that can be generated by this method etc.
> >
> > **Discussion on limitations**: Thank you for pointing this out. Indeed, periodicity is not supported by the proposed model, although extending it to accommodate periodicity is an interesting avenue. Additionally, without detailed empirical investigations, it is uncertain how well ModFlow would perform in the context of larger biomolecules ($\geq$ 10K atoms). We will add a discussion elaborating on these limitations in the revised version.
> >
> > > Also, potential negative societal impact (such as generating toxic materials, etc.) should be mentioned.
> >
> > **Discussion about negative societal impact** Thank you for this important suggestion. Indeed, despite our best intentions, we're not in a position to guard against the dangers that the method proposed in negative societal impact this work could pose when used as is, or adapted to, harm society, e.g., by generating extremely potent chemical weapons such as nerve agents. As you pointed out, there is growing evidence [1] about how generative models, similar to the ones we consider here, could facilitate the design of such chemicals. We very much understand and will acknowledge such perils in the final version.
> >
> > [1] F. Urbina, F. Lentzos, C. Invernizzi, and S. Ekins. Dual use of artificial-intelligence-powered drug discovery, Nature Machine Intelligence, 4(3):189-191, 2022.
> >
> >
> > Thank you very much for your thoughtful comments and suggestions, which have led to more comprehensive empirical evidence in support of the proposed model. We hope that your concerns have been sufficiently addressed, and if so, you would consider raising your score.

---

> > > ### Comment · Reviewer_c5ye · 2022-08-07
> > > **Updated review**
> > >
> > > I thank the authors for their detailed response. The runtime performance & scalability looks good. I'm not fully convinced that the new evaluation metrics are sufficient for real world applications, but the fact that the proposed method outperforms existing methods across metrics adds more confidence in the method.
> > >
> > > I will update my rating to 6.

---

> > > > ### Author Response · Authors · 2022-08-08
> > > > **Reply to reviewer comment**
> > > >
> > > >  Many thanks for your response! We’re glad that the additional evaluation reinforced your confidence in the proposed method, and grateful for your acknowledgment in terms of a revised score.

---

### Official Review · Reviewer_FLBL · 2022-07-12

**Rating:** 5
**Confidence:** 2
**Soundness:** 2 fair
**Presentation:** 2 fair
**Contribution:** 2 fair

**Summary:**

The presented method combines continuous E(3)-equivariant flows and PDEs on graphs to generate molecular structures.

**Questions:**

- Eq. (1) seems to encapsulate a central assumption of this probabilistic model. Why should it be that, given edge information, the nodes can be labeled independently?
- Sometimes, I am unsure whether you are working with 2D or 3D graphs. On the one hand, you use concepts such as bond orders and chemical validity; on the other, you are working with 3D structures. Do you convert generated 3D structures back to 2D graphs?
- Tree representation:
  - Is it always possible to decompose a molecular graph into a tree-like structure?
  - I understand that the model generates 3D molecular structures. How do you handle substructures, i.e., groups of atoms consisting of more than one atom?
- How is the neighborhood of an atom $\mathcal{N}_i$ defined? Do you recompute that throughout the generation process?
- Section 3.3: are you confusing invariant and equivariant in this section?

**Limitations:**

The authors adequately addressed the limitations of their work.

**Strengths And Weaknesses:**

# Strengths

- Figure 2 helped me get a high-level understanding of the model
- Experiment 4.1 show-cased the capabilities of the model with a simple example
- Sections 4.2 and 4.3 are easy to follow and fairly clear.
- The results seem impressive: the model's scores on "validity", "uniqueness", "novelty", and "reconstruction" seems impressive.

# Weaknesses

- The paper is not exceptionally well-written. I had a hard time following Sections 3.2 and 3.3 -- the main contributions of this paper.
- The reader is expected to be familiar with the EGNN model. If the reader doesn't happen to know the exact inputs and mode of operation of this model, the reader will get lost. The importance of this borrowed architecture in this approach warrants a short introduction.

---

> ### Author Response · Authors · 2022-08-02
> **Reply to the comments and suggestions**
>
> Many thanks for your thoughtful feedback!
>
> >The paper is not exceptionally well-written. I had a hard time following Sections 3.2 and 3.3 -- the main contributions of this paper.
>
> We apologize for this. Certainly writing for these parts could be improved, and we'll indeed refine the descriptions based on your feedback. Part of this confusion, we believe, stems from the many subtleties underlying the proposed model, notably, but not restricted to,
> designing a _continuous_ flow pipeline that requires a delicate interplay of multiple methodologies, ranging from dynamical systems to graphs, for treading a complex _discrete_ space.
>
> To paraphrase, we associate a latent score variable to each node of the graph, which represents the probability of that node having a certain label (e.g. carbon, oxygen, etc). In Section 3.2 we present a continuous normalizing flow based on neural ODEs (Chen 2018), which allows the latent scores to evolve guided by a neural network $f_\theta$. The ODE aspect of the flow lets the latent scores evolve continuously, without encountering ``jumps" or discontinuities that happen in autoregressive or recurrent models. Here a key contribution is that each node in the graph can only look at neighboring nodes to decide how it can evolve, and this induces a local inductive bias (analogous to Laplacian) in the atom assignment. The ODE flow is a known technique, however extending it for graph generation - which involves coupled ODEs (i.e., individual ODE flows, one per node, influenced by neighboring nodes) - is non-trivial; and to our knowledge, this work is the first such generative model for graphs.
>
> In 3.3 we describe desiderata for the flow function. The coupled flow should be ``equivariant" to permutations; and additionally equivariant to rotations, reflections, and translations when we have spatial 2D/3D coordinates for the nodes. That is,, doing such operations on a graph that represents a molecule should not fundamentally change the flow. This ensures that the likelihood of the molecule is preserved under the model if instead of a molecule graph, we provide the model a transformed graph as input such as the one obtained by permuting (reordering) the nodes, or rotating it by any angle when the nodes have 2D/3D positions assigned to them. We use a known EGNN method to define $f_\theta$, in order to serve this purpose. While EGNNs have been explored in prior art, their use in our setting turns the proposed model, ModFlow, into a special temporal graph network. We show this connection formally in the Appendix.
>
> We emphasize that a major contributions of this method is a physics-inspired co-evolving continuous-time flow that induces useful inductive biases for a highly complex combinatorial setting, and thus affords effective sampling from a distribution that closely approximates the true underlying distribution. This is evident from strong empirical performance of the proposed methods across many metrics, including an exceptionally high generative molecule validity.
>
> Thanks for your suggestion - we will add a short introduction to make Sections 3.2 and 3.3 more accessible.
>
> > The reader is expected to be familiar with the EGNN model. If the reader doesn't happen to know the exact inputs and mode of operation of this model, the reader will get lost. The importance of this borrowed architecture in this approach warrants a short introduction.
>
> Thanks for this excellent suggestion. We agree that the EGNN is cursorily presented, and a longer exposition would be useful. **We have added a short tutorial of EGNNs in the Appendix for this purpose**. In the final version, we will elucidate that the need for EGNN arises from the normalizing flow, where the probabilities evolve as a function of the nodes and their neighbors, by definition (Eq 3). We would also underscore that our extensive ablation studies reveal that  incorporating equivariance, using EGNN, improved performance by roughly 5%-units (90% to 95% validity on ZINC, 93% to 98% validity on QM9)
>
> >Eq. (1) seems to encapsulate a central assumption of this probabilistic model. Why should it be that, given edge information, the nodes can be labeled independently?
>
> This is an excellent observation, and we agree this can be misleading. Eq 1 defines the likelihood, where indeed each node is labelled independent of its neighbors, conditioned on latent variables $z_i$. However, the latent variables $z_i,z_j,...$ are dependent, resulting in dependencies between atom assignments. We will clarify this in the paper, and also use a more precise notation of $p(V|E, {z })$ to emphasize the (previously missing) conditioning.

---

> > ### Author Response · Authors · 2022-08-02
> > **Reply to the comments and suggestions**
> >
> > > Sometimes, I am unsure whether you are working with 2D or 3D graphs. On the one hand, you use concepts such as bond orders and chemical validity; on the other, you are working with 3D structures. Do you convert generated 3D structures back to 2D graphs?
> >
> > Our model applies equally to coordinate-free graphs, and to 3D or 2D (planar) graphs. If angle or coordinate information is available, it is provided as an extra input to the function $f_\theta$, along with the latent scores $z_i$. Absent this information, $f_\theta$ accepts the latent scores $z_i$ as its input. In either case, the core of the generative process is governed by the latent scores of the neighbors. We will clarify this by including the 3D/2D information formally in Eq 3. In our experiments, we obtain 3D graphs from SMILES using RDKit.
> >
> > > Tree representation:
> > - Is it always possible to decompose a molecular graph into a tree-like structure?\\
> > - I understand that the model generates 3D molecular structures. How do you handle substructures, i.e., groups of atoms consisting of more than one atom?
> >
> > Yes, it is always possible to represent a molecular graph as a junction tree using an appropriate vocabulary, as elucidated in Jin et al: “Junction Tree Variational Autoencoder for Molecular Graph Generation” (ICML’18). Substructures of interest, e.g., rings, are included in the vocabulary (i.e., the vocabulary consists of both atoms and these larger substructures). The flow evolves on the junction tree, and is subsequently used to label each node in the junction tree with an atom or a group of atoms. As a result, the junction tree can be expanded back to a molecular graph with possibly complex substructures.
> >
> > > How is the neighborhood of an atom  defined? Do you recompute that throughout the generation process?
> >
> > We use a simple 1-hop neighborhood, where the neighbors of a node (atom/substructure) are the nodes that it shares a bond (edge) with.  The neighborhood is fixed at the beginning (according to the junction tree) and stays same throughout the generation process.
> >
> > > Section 3.3: are you confusing invariant and equivariant in this section?
> >
> > Thank you for the opportunity to clarify this. Our interests are in achieving equivariance, i.e., transforming inputs should result in an equivalent transformation of outputs. Invariance often arises a special case of equivariance, where the transformation in the output space is an identity map, i.e., the output does not change with change in the input. For example, invariance results when a neural network implements a composition of equivariant layers followed by an invariant layer. We've updated section 3.3 to focus solely on equivariance for improved readability.
> >
> > Thank you very much for your constructive feedback! We hope we've sufficiently addressed your concerns, and if so, would appreciate a revised score.

---

> > > ### Author Response · Authors · 2022-08-09
> > > **Reply to reviewer comment**
> > >
> > > Many thanks, again, for your feedback that provided us with an opportunity to clarify some important aspects of the proposed method. We would greatly appreciate your stronger support for this work.

---

### Official Review · Reviewer_obfC · 2022-07-13

**Rating:** 5
**Confidence:** 4
**Soundness:** 3 good
**Presentation:** 3 good
**Contribution:** 3 good

**Summary:**

The authors of this paper propose to use a continuous-time normalizing flow and E(3)-Equivariant GNN as the differential function in the neural PDE for molecular generation. Experimental results on QM9 and ZINC show that the proposed model, ModFlow, obtained better validity and novelty in comparison with existing graph flow based models.

**Questions:**

The authors should (1) use more informative metrics, (2) dissect a test set for each dataset, and (3) compare with non-flow-based state of the art molecular generation models.

**Ethics Review Area:**

["I don’t know"]

**Limitations:**

Only technical limitations are mentioned as future work in one sentence in Section 5 Conclusion.

**Strengths And Weaknesses:**

Strengths:
1. The consideration of using continuous-time normalizing flow as encoder makes sense. The authors find the their proposed model is equivalent to a variant of temporal graph network (TGN).

Weakness:
1. The evaluation metrics used in this paper are validity, uniqueness, novelty, and reconstruction, which are usually used for sanity check. To further look into the consistency of between the generated samples and the training samples, more informative metrics should be used. For example, the metrics in MOSES can be considered.
2. It seems that an intact test set was not used for result evaluation.
3. Furthermore, the computational comparison should not be limited to flow-based methods. Other state-of-the-art deep learning based molecular generation approaches should be compared as well.
4. Minors: (1) Moroever -> Moreover. (2) Check the capitalizations in each reference item.

---

> ### Author Response · Authors · 2022-08-02
> **Reply to the comments and suggestions**
>
> Many thanks for your insightful feedback! We are grateful for your suggestions and have implemented all of them to address
> your concerns, as we describe below in detail.
>
> > The evaluation metrics used in this paper are validity, uniqueness, novelty, and reconstruction, which are usually used for sanity check. To further look into the consistency of between the generated samples and the training samples, more informative metrics should be used. For example, the metrics in MOSES can be considered.
>
> **MOSES metrics**:  We agree that while novelty, validity, uniqueness, and reconstruction (NVUR) are common in literature,
> stronger metrics, such as MOSES, provide more nuanced evidence. Based on your feedback, we report performance with
> respect to the four MOSES metrics which are
> - **Frechet Chemnet distance (FCD)**: a general purpose metric that measures diversity as well as chemical and biological property alignment between generated and reference molecules. _Lower is better_
> - **Frag**: looks at BRICS fragment matching between generated and reference molecules. _Higher is better_
> - **Similarity to nearest neighbor (SSN)**: looks at closest distance between generated and reference molecules, and quantifies how close the generated molecules are to the true molecule manifold. _Higher is better_
> - **Internal diversity (IntDiv)**: accounts for diversity of the generated set._Higher is better_.
>
> We computed the above metrics for the proposed method (ModFlow) and 5 state-of-the-art methods including both flow-based (GraphAF, GraphDF, MoFlow) and non-flow (GVAE, JTVAE, GraphEBM) methods. **ModFlow achieves best results across these metrics on both the ZINC250K and QM9 datasets** as shown in the tables below. For instance, on QM9, the scores are $\approx$ (0.40 vs 0.50 for best competing) on FCD, (0.93 vs 0.89 best competing) on Frag, (0.62 vs 0.58 best competing) on SSN, and (0.88 vs 0.85 best competing) on IntDiv.  Notably, ModFlow registers a lower FCD score and higher IntDiv value compared to other methods, which suggests it is able to generate diverse set of molecules similar to those present in dataset
>
> We will include a discussion about the new metrics, and add these results as a table in the final version.
>
> For QM9:
> |Method|FCD|Frag|SNN|IntDiv|
> |-----|----|-----|-----|------|
> |GVAE|0.513|0.821|0.582|0.822|
> |GraphEBM  |0.551|0.831|0.547|0.831|
> |GraphAF|0.732 | 0.863| 0.565 | 0.823|
> |GraphDF|0.683 |0.892|0.562 |0.839|
> |MoFlow|0.496 | 0.840 |0.502 |  0.852|
> |ModFlow(2D-EGNN)|0.432| 0.928 | 0.608 | 0.875|
> |ModFlow(3D-EGNN)|0.478| 0.934 | 0.613 | 0.885|
> |ModFlow(JT-2D-EGNN)| 0.421| 0.921| 0.595 | 0.867|
> |ModFlow(JT-3D-EGNN)|**0.401** | **0.939** | **0.624** | **0.889**
>
> For ZINC:
> |Method|FCD|Frag|SNN|IntDiv|
> |-----|----|-----|-----|------|
> |JTVAE|0.512|0.890|0.5477|0.855|
> |GVAE|0.571|0.871|0.532|0.852|
> |GraphEBM|0.613|0.843|0.487|0.821|
> |GraphAF|0.524|0.803|0.465|0.855|
> |GraphDF|0.658|0.869|0.515|0.829|
> |MoFlow|0.597|0.851|0.452|0.832|
> |ModFlow(2D-EGNN)|**0.495** | 0.891 | 0.570 | 0.863|
> |ModFlow(3D-EGNN)|0.512| 0.905 | 0.584 | 0.869|
> |ModFlow(JT-2D-EGNN)| 0.501| 0.915 | 0.563 | 0.857|
> |ModFlow(JT-3D-EGNN)|0.523| **0.929** | **0.594** | **0.879**|
> >It seems that an intact test set was not used for result evaluation.
>
> Based on your feedback, we have performed analysis over a holdout reference dataset, and compared generated molecules from our model against these reference compounds. For our purpose, we evaluated the generated structures via distributions of their important properties. Specifically, we obtained kernel density estimates of these distributions to aid in visualization. We have used
> - **Molecular weight**: sum of the individual atomic weights of a molecule. Provides insight about the bias of the generated molecules toward lighter or heavier molecules.
> - **logP**: ratio of concentration in octanol-phase to aqueous phase, also known as the octanol-water partition coefficient. It is computed via Crippen [1] estimation.
> - **Synthetic Accessibility Score (SA)**: an estimate for the synthesizability of a given molecule. It is calculated using the molecule's fragments contribution [2].
> - **Quantitative Estimation of Drug-likeness (QED)**: a value describing likeliness of a molecule as a viable candidate for a drug. It ranges between [0,1] and captures the abstract notion of aesthetics in medicinal chemistry [3].
>
> **Our experiments reveal that ModFlow generally matches the distributions well, barring some discrepancy in QED and logP, demonstrating its effectiveness in generating molecules similar to those in the reference set.** In particular, the generated molecules have similar distributions with respect to MW and SA. We provide more details and visualizations in the Supplementary of the revised version.
>  Links to Plots: https://postimg.cc/Q9Gq6LGH , https://postimg.cc/phfJ40Rk

---

> > ### Author Response · Authors · 2022-08-02
> > **Reply to Reply to the comments and suggestions**
> >
> > > Furthermore, the computational comparison should not be limited to flow-based methods. Other state-of-the-art deep learning based molecular generation approaches should be compared as well.
> >
> > **Comparison with state-of-the-art non-flow based methods.** Based on your suggestion, we compared our method with three state-of-the-art non-flow based methods as well: Grammer Variational Autoencoder (GVAE) [4], MolecularRNN (MRNN) [5], and Graph Convolutional Policy Network  (GCPN) [6]. ModFlow significantly outperforms all these methods on the Zinc250K data as shown in the table below: MRNN (65%) achieves 65% validity, while the other two methods - GVAE and GCPN - manage only 20% validity, compared to about 95-98% for ModFlow. We have included these results in the revised version, and will also add the comparisons on the QM9 data.
> > For ZINC:
> > |Method| Validity %| Uniqueness %| Novelty %| Reconstruction %|
> > |--------|---------|-----------|-----------|---------|
> > | **(Non-flow based methods)**|
> > |MRNN|65 | 99.89  |  100  | n/a |
> > |GVAE | 7.2 |9 | 100 |53.7|
> > |GCPN |20|  99.97 |100 |n/a|
> > |**(Flow based methods)**|
> > |GraphNVP|42.6 | 94.8  |  100  |100|
> > |GRF | 73.4 | 53.7  |  100 |100|
> > |GraphAF|68 | 99.1  |  100   | 100|
> > |GraphDF | 89 | 99.2 | 100  | 100|
> > |MoFlow| 50.3|**99.9** |100 |100|
> > |ModFlow (2D-EGNN) |**94.8 $\pm$ 1.0**| 99.4 | 100 |100|
> > |ModFlow (3D-EGNN) |**95.4 $\pm$ 1.2**| 99.7 | 100 |100|
> > |ModFlow (JT-2D-EGNN) |**97.4 $\pm$ 1.4**| 99.1 | 100 |100|
> > |ModFlow (JT-3D-EGNN) |**98.1 $\pm$ 0.9**| 99.3 | 100 |100|
> >
> > > Only technical limitations are mentioned as future work in one sentence in Section 5 Conclusion.
> >
> > **Non-technical ramifications**: thank you for bringing our attention to some important non-technical limitations. Indeed, despite our best intentions, we're not in a position to guard against the dangers that the method proposed in this work could pose when used as is, or adapted to, harm the society, e.g., by generating extremely potent chemical weapons such as nerve agents. In fact, there is growing evidence [7] about how generative models, similar to the ones we consider here, could facilitate the design of such chemicals. We very much understand and will acknowledge these perils in the final version, and hope to be able to assist policy-makers in mitigating such risks to the best of our abilities.
> >
> > >The authors should (1) use more informative metrics, (2) dissect a test set for each dataset, and (3) compare with non-flow-based state of the art molecular generation models.
> >
> > We thank you, again, for your great suggestions (and for catching minor details, which we have fixed now). The potential of the proposed method has been significantly reinforced with these meaningful evaluations, and we hope the same is reflected in your revised scores.
> >
> >
> > [1] S. Wildman and G. Crippen. Prediction of physicochemical parameters by atomic contributions, Journal of chemical information and computer sciences, 39(5):868–873, 1999
> >
> > [2] P. Ertl and A. Schuffenhauer. Estimation of synthetic accessibility score of drug-like molecules based on molecular complexity and fragment contributions, Journal of cheminformatics, 1(1):1–11, 2009
> >
> > [3] G. Bickerton, G. Paolini, J. Besnard, S. Muresan, and A. Hopkins. Quantifying the chemical beauty of drugs, Nature chemistry, 4(2):90–98, 2012.
> >
> > [4] M. Kusner, B. Paige, and J.  Hernández-Lobato. Grammar variational autoencoder. In ICML, 2017. \\
> >
> > [5] M. Popova, M. Shvets, J. Oliva, and O. Isayev. MolecularRNN: Generating realistic molecular graphs with optimized properties  arXiv:1905.13372, 2019.
> >
> > [6] J. You., B. Liu, Z. Ying, V. Pande, and J. Leskovec. Graph convolutional policy network for goal-directed molecular graph generation. In NeurIPS, 2018
> >
> > [7] F. Urbina, F. Lentzos, C. Invernizzi, and S. Ekins. Dual use of artificial-intelligence-powered drug discovery, Nature Machine Intelligence 4(3):189-191, 2022.

---

> > > ### Author Response · Authors · 2022-08-09
> > > **Reply to reviewers comments**
> > >
> > > Many thanks, again, for your constructive feedback! We believe the efficacy of the proposed method has been further strengthened with strong empirical evidence on each of the evaluation you suggested (MOSES metrics, test set and non-flow methods) . We would be grateful if you could acknowledge the same, and revise your scores.

---

> > > > ### Comment · Reviewer_obfC · 2022-08-09
> > > > **Most of my questions are answered**
> > > >
> > > > I have read the authors' rebuttals and saw that this work has been improved by including more metrics, comparisons with non-flow-based generative methods. Moreover, computing time and molecular optimization were also reported. Thus, I would like to change my rating to borderline accept. For further improvement, the authors should include density distributions of other methods as well to the property-wise distribution plots.

---

### Author Response · Authors · 2022-08-02
**A summary of updates**

We thank the reviewers for their insightful comments and suggestions to improve this work, and to the area, program, and general chairs for their great service to the community. The reviewers raised many good points, especially regarding stronger evaluation metrics, but also questions about scalability and runtime, clarity, more comprehensive comparisons, and property optimization. We summarize their main concerns and questions, and detail the additional analyses and experiments conducted by us to address these concerns. We will include all the additional analyses and experiments in the final version.

**MOSES metrics, distribution of properties relative to a large hold-out set, molecular optimization, etc.**

- Three reviewers raised concerns about the metrics used previously (novelty, uniqueness, validity, reconstruction). As they suggested, we have added four stronger metrics pertaining to the MOSES benchmark: (1) fragment similarity (**Frag**), (2) nearest neighbor similarity (**SNN**), (3) Frechet Chemnet distance (**FCD**), and (4) Internal Diversity (**IntDiv**).
- We computed the above metrics for ModFlow and several other state-of-the-art baselines. Notably, ModFlow achieves best results on all four metrics: FCD ( 0.40 vs 0.50 for best competing), in Frag ( 0.93 vs 0.89 best competing), in SSN ( 0.62 vs 0.58 best competing), and in diversity ( 0.88 vs 0.85 best competing).
- We additionally compare property distributions between molecules generated by ModFlow and a large hold-out reference dataset of molecules. We examined molecular weight, synthetic accessibility score (SAS), QED, and logP as chemical properties. ModFlow generated molecules that were found to have largely similar property distributions to the reference compounds, barring some discrepancy in the QED.

**Runtime and Scalability**

Based on reviewers' suggestion to include a discussion over training and inference time, we have included a discussion on both the training time and generation time of ModFlow and other models. We also compared several generation speed across methods. Notably, ModFlow turns out the fastest generator out of the competing methods with around 0.50 secs per molecule, compared to 0.70-3.1 secs for other methods (on ZINC250K).

**More state-of-the-art baselines, including the non-flow based methods**

In addition to the five flow-based competing methods, we have also added three more competing methods to our result tables (namely, MRNN, GVAE and GCPN). ModFlow outperforms them significantly.

**Property Optimization**

Per reviewers' suggestion, we also performed property-targeted molecule optimization, which aims to search for molecules with better scores for a chemical property. We used QED score as our target chemical property to interpolate in the latent space, and discovered new molecules with higher QED. ModFlow is able to find novel molecules with high QED scores.  We report these novel molecules, sorted by their QED scores.

We also address all specific questions, comments, and concerns raised  by reviewers in our detailed individual response for each reviewer.

---

### Meta-Review · Area_Chair_gFE1 · 2022-08-29

**Recommendation:** Accept
**Confidence:** Certain

**Metareview:**

This paper introduces ModFlow, an E(3) equivariant normalizing flow for generating molecular conformations. A normalizing flow model for molecular systems could be extremely useful since normalizing flows have desirable properties including tractable densities and good sampling behavior. The authors demonstrated the effectiveness of their method compared to state of the art on two canonical datasets (QM9 and ZINC250K) although it might have been nice to see something harder.

All four referees (weakly) thought the paper should be accepted. All of the referees liked the idea of using normalizing flows for this problem and thought the approach laid out by the authors seemed reasonable. Some of the referees thought the writing was good, while others thought it could be improved. The main stumbling block was the choice of metrics reported on the QM9 and ZINC datasets. However, during the rebuttal the authors provided additional metrics which seemed to corroborate the success of the approach. Given the global support by the referees and improvement by the authors during the rebuttal, I think the paper ought to be accepted.


**Award:**

No

---

### Decision · Program_Chairs · 2022-09-14

Accept